# Sex Hormone-Specific Neuroanatomy of Takotsubo Syndrome: Is the Insular Cortex a Moderator?

**DOI:** 10.3390/biom12010110

**Published:** 2022-01-10

**Authors:** Michiaki Nagai, Carola Yvette Förster, Keigo Dote

**Affiliations:** 1Department of Cardiology, Hiroshima City Asa Hospital, Hiroshima 731-0293, Japan; k-dote@asa-hosp.city.hiroshima.jp; 2Department of Anaesthesiology, Intensive Care, Emergency and Pain Medicine, University of Würzburg, D-97080 Würzburg, Germany; Foerster_C@ukw.de

**Keywords:** insular cortex, autonomic nervous system, laterality, sex hormone, central autonomic network, Takotsubo cardiomyopathy

## Abstract

Takotsubo syndrome (TTS), a transient form of dysfunction in the heart’s left ventricle, occurs predominantly in postmenopausal women who have emotional stress. Earlier studies support the concept that the human circulatory system is modulated by a cortical network (consisting of the anterior cingulate gyrus, amygdala, and insular cortex (Ic)) that plays a pivotal role in the central autonomic nervous system in relation to emotional stressors. The Ic plays a crucial role in the sympathovagal balance, and decreased levels of female sex hormones have been speculated to change functional cerebral asymmetry, with a possible link to autonomic instability. In this review, we focus on the Ic as an important moderator of the human brain–heart axis in association with sex hormones. We also summarize the current knowledge regarding the sex-specific neuroanatomy in TTS.

## 1. Introduction

Takotsubo syndrome (TTS)—a transient form of left ventricular (LV) dysfunction in individuals who are under emotional stress—has been known as an acute syndrome with akinesis from the heart’s mid-left ventricle to the apex [1]. Several pathophysiologic conditions are thought to be linked in the brain–heart regulatory axis and are associated with left ventricular (LV) dysfunction. The pathophysiology that underlies the dysregulation of cardiovascular function by the central autonomic nervous system has been extensively investigated [2]. It has been speculated that a cortical network comprised of the anterior cingulate cortex (ACC), amygdala, and insular cortex (Ic) regulates the human cardiovascular system, and that the network of these cortical regions is necessary to regulate the central autonomic network (CAN) in response to emotional stress [3]. Clinical studies revealed that damage involving the Ic is associated with QT prolongation, myocardial injury, and a higher plasma level of catecholamine; these conditions have also been observed in individuals with TTS [4].

Studies of gender-specific responses to changes in sympathovagal balance and asymmetry in cerebral function have suggested that a dominant pattern in females is determined by the left hemisphere, since functional cerebral asymmetry appears to be changed according to the body’s estrogen levels. The sympathovagal autonomic balance might be shifted via changes in CAN outflow due to functional cerebral asymmetry variability [5].

The relationship between TTS and the Ic has been reviewed [4]. The present review describes the sex-specific neuroanatomy in TTS and updates the knowledge regarding the integrated roles of the Ic in the brain–heart axis. We discuss these topics mainly from a pathophysiological standpoint, focusing on the relationships between the Ic and sex hormones and TTS. Finally, we summarize the current insights in the field of neuroanatomy regarding the relationship between Ic damage and TTS, which is speculated to be moderated by estrogen deficiency.

## 2. The Brain–Heart Connection

### 2.1. Cortical Regulation in the Circulatory System in Response to Emotional Stress

#### The Central Autonomic Network (CAN)

The CAN is composed of the periaqueductal gray matter, parabrachial nucleus (PBN), nucleus tractus solitarius (NTS), ventrolateral medulla (VLM), hypothalamus, amygdala, and the Ic (Figure 1). The CAN functions as an integral component of an internal regulation system with which the brain controls the visceromotor and neuroendocrine systems [4,6]. The involvement of a network within the CAN consisting of the amygdala, ACC, and Ic was shown to be associated with the relationship between the processing of emotional information and autonomic nervous system responses [7].

### 2.2. The Insular Cortex and the Processing of Emotion

Several neuroimaging studies revealed that activation of the Ic is involved in the processing of negative emotions such as anger, fear, and anxiety [7]. In an investigation using positron emission tomography (PET), elevated regional cerebral blood flow (rCBF) in the Ic was demonstrated during recall-generated sadness in healthy subjects [8,9], suggesting that the Ic participates in the emotional response to dangerous or distressing situations [9]. In another PET study [10], activation of the Ic was observed to be associated with the processing of anger, fear, happiness, and sadness. Panic attacks induced by lactate were shown to be associated with significantly increased rCBF in the Ic of patients with panic disorder [11]. Later research indicated that the anterior Ic is critical for emotional awareness [12].

The Ic is thus suspected to play an important role in coordinating the autonomic nervous system with the information of emotion [3]. The above-described findings confirm the hypothesis that activation of the anterior Ic might shift the predominant neutral interaction of emotional awareness from cardiovascular regulation to a sympathetic-predominant status.

### 2.3. The Insular Cortex and the Regulation of the Cardiovascular System

A.Animal Studies

A study using anesthetized rats revealed the cardiac chronotropic sites. While stimulations in rostral posterior Ic derived tachycardia, those in the caudal posterior Ic derived bradycardia [13,14]. In another study using rats, stimulation of the Ic was associated with atrioventricular block, QT prolongation, asystole, and death, along with an elevation in the plasma norepinephrine level as well as myocardial cytolysis [15]. In a rat model of electrical kindling, Ming et al. [16] implanted electrodes in the granular layer of the Ic, and in 92.3% of the rats, a seizure was induced. In older rats, the stimulation was associated with a greater heart rate (HR) reduction, and the 25% of the kindled rats died.

In another investigation using rats, ischemic stroke in Ic was associated with left atrium (LA)-pulmonary vein (PV) border fibrosis, which is thought to be the pathophysiology of the generation of atrial fibrillation (AF) [17]. LA-PV border fibrosis due to experimental ischemic stroke might trigger local inflammation via preganglionic fibers ending in ganglionated plexi [17].

In a model of permanent middle cerebral artery occlusion (MCAO), the severity of left Ic lesion and the serum and heart norepinephrine levels were significantly higher in MCAO mice with LV dysfunction than those with normal cardiac function [18]. In another murine model, significantly increased expressions of p-p38 MAPK, bax, and cleaved caspase-3 occurred in kidney tissue in a right-side transient MCAO group compared to the sham-operated group; transient MCAO resulted in sympathetic nervous system hyperactivity, kidney dysfunction, upregulations of apoptosis in kidney tissue, and renal fibrosis in the mice [19].

The cardiovascular, autonomic, and cardiac changes produced in a focal experimental model of intracerebral hemorrhage (ICH) in the Ic were evaluated by Marins and colleagues [20]. Although the ICH increased the rats’ baseline HR, cardiac ectopies were markedly increased after 24 h of ICH specifically in the right Ic. Moreover, only in the right Ic ICH cases, the rats hearts’ weight, calcium amplitude, and sarco/endoplasmic reticulum Ca2+-ATPase expression were reduced [20].

Although the type of Ic ischemia or ICH model might have differing impacts on the target organs in these animal studies, these studies’ findings reveal that Ic stroke is associated with arrhythmia, LV dysfunction, LA-PV border fibrosis, and kidney injury. Because there were few observations from the viewpoint of gender differences, in the future, it would be necessary to examine the gender-specific brain–heart relationship in these animal experiments.

B.Human Studies
B.1.The Role of Hemispheric Laterality in Sympathovagal Modulation


In epileptic patients who were undergoing a preoperative evaluation, the HR decreased after an amobarbital injection to the right carotid artery that inactivated the right hemisphere, whereas the HR increased after an amobarbital injection to the left carotid artery that inactivated the left hemisphere [21]. In their analysis of heart rate variability (HRV), Yoon et al. [22] observed that the low frequency (LF) per high frequency (HF) ratio, which represents the activity of the sympathetic nervous system, significantly changed with the left hemisphere inactivation. In a more recent analysis of 15 drug-refractory epileptic patients, inactivation of the right hemisphere significantly increased the HF component of blood pressure (BP) and HR, which are indicators of parasympathetic nervous system activity, and inactivation of the left hemisphere increased the LF component of BP and HR, which are indicators of parasympathetic nervous system activity [23].

These findings suggest that the left and right hemisphere might have different cerebral impacts on autonomic function: the right side of the brain is involved in sympathetic cardiac function, whereas the left side of brain is involved in parasympathetic cardiac function.

B.2.The Insular Cortex and Heart Rate, Arrhythmia, and Sympathovagal BalanceB.2.1.Epilepsy Studies

Neuroanatomic brain–heart connections are involved in cardiac arrhythmias that occur in response to abnormal brain activation including the Ic. Seizures induce a variety of transient cardiac effects such as arrhythmias, asystole, and other electrocardiography (ECG) abnormalities that can trigger sudden unexpected death in individuals with epilepsy [24]. For example, cardiac asystole has been reported in patients with partial seizures involving the left Ic [25,26,27] as well as those involving the right posterior Ic [28].

Focal cortical dysplasia in the temporal region including the Ic is suggested to cause ictal bradycardia and ictal asystole. A 15-year-old epileptic patient with focal dysplasia of the cerebral cortex that included both the right and left Ic showed prolonged asystole in relation to seizures and syncope [29]. In addition, a 25-year-old patient revealed asystole induced by Ic seizures, which was confirmed based on intracerebral electroencephalography. A recent investigation using hybrid PET-magnetic resonance imaging (MRI) revealed hypometabolism in the posterior Ic in the right hemisphere and a fluid-attenuated inversion recovery hyperintensity in the same Ic region [30]. In a whole-brain statistical analysis of PET images, hypometabolism in the right posterior Ic was shown in epileptic patients with ictal asystole compared to healthy subjects and epileptic patients without ictal asystole [31].

Stimulation of the Ic prior to a lobectomy for the purpose of seizure control has been performed in patients with epilepsy; while depressor and bradycardia responses were frequently observed when left Ic stimulation was applied, pressor and tachycardia responses were frequently observed when right Ic stimulation was applied [32]. In contrast, a chronotropic and inotropic depressor function was reported in relation to the right and left Ic, and stimulation of both sides of the Ic induced significant decreases in cardiac output and HR, plus a significant increase in stroke volume without differences between the right and left hemispheres [33].

In a study that applied 100 electrical stimulations to the Ic in 47 patients with epilepsy, 26 stimulations (26%) induced bradycardia and 21 stimulations (21%) induced tachycardia [34]. Bradycardia and tachycardia were induced by right Ic stimulations as often as by left Ic stimulations. The LF/HF increase was accompanied by tachycardia, whereas the HF increase was accompanied by bradycardia. Some asymmetry of the left/right side of Ic subregions was observed with the increased or decreased HR after stimulation. However, there was a spatial tachycardia distribution predominantly in the posterior part of the Ic, whereas a spatial bradycardia distribution was observed in a more anterior part of the median Ic [34].

A study using stereo-electroencephalography (SEEG) demonstrated that a left amygdalo-hippocampal seizure induced asystole with ipsilateral spread to the Ic [35]. Although the cardiac parasympathetic nervous system had been suggested to be regulated by the left Ic, the bilateral Ic in relation to bradycardia was reported by a recent SEEG study of Ic stimulation [34]. Indeed, asystole in patients with epilepsy has been reported for both right and left hemispheric lesions in Ic [36]. It is unclear whether the unilateral Ic involvement is sufficient to evoke asystole, because a non-negligible propagation from the right Ic to the left Ic was observed [31]. In an earlier report about an animal model, transcallosal reciprocal insulo-insular connectivity was reported to play an important role in the sympathovagal balance in cardiovascular regulation [37].

B.2.2.Neuroimaging Studies

The activation of interconnected regions of the brainstem, ACC, amygdala, and Ic has been shown to predict HR acceleration in relation to emotional facial stimuli [38]. In a study using combined HRV-functional MRI (fMRI), HF power was significantly correlated with activity in the PBN, locus coeruleus, periaqueductal gray, hypothalamus, amygdala, and posterior Ic [39]. Because significant correlations were observed between activation of the left Ic and HF in other studies, the left Ic is considered to be predominantly associated with parasympathetic autonomic nervous system [40,41,42].

Within the whole cortex, the highest α4β2 nicotinic acetylcholine receptor (nAChRs) densities are in the anterior Ic and the dorsal ACC. A PET study investigated the association between parasympathetic tone as assessed by HRV and the regional brain availability of nAChR assessed by the potential of 18F-F-A-85380 binding; higher availability of nAChRs in the right dorsal anterior Ic and the bilateral dorsal ACC was associated with lower HF power. The increased nAChR availability in the right anterior Ic was suggested to participate in reduced parasympathetic activity [43]. In a voxel-based lesion symptom mapping study, a left posterior Ic lesion cluster had a significant predominant association with an increased LF/HF ratio, indicating that the sympathetic nervous system activity was increased according to the left Ic lesions [44].

In another fMRI study, the increased activity in the dorsal ACC and right anterior and posterior Ic plus the decreased activity in the ventral ACC and left posterior Ic had significant correlations with muscle sympathetic nerve activity burst frequency [45]. In a large fMRI study of 310 community-dwelling adults, a standardized battery of stressor tasks was performed. A larger BP rise induced by the task covaried positively with right anterior Ic activity, while that rise covaried negatively with activation of the left anterior Ic [46].

Song et al. [47] investigated the neural responses to the Valsalva maneuver in 29 heart failure patients with New York Heart Association Functional Class II and 35 healthy controls by conducting a blood oxygen level-dependent fMRI study. The heart failure patients exhibited decreased neural activation in the cerebellum vermis, cortices, bilateral postcentral gyrus, left putamen, and left Ic. The magnitude of fMRI responses in the cerebellum and Ic among the heart failure patients was decreased or the responses were delayed in accord with the Valsalva maneuver. The severity of tissue changes was associated with the impairment of functional responses in cerebellar sites and the Ic [47]. These results indicate that the exaggerated sympathetic tone in heart failure might be gated by the decreased neural activation in the left Ic that is associated with the activity of the parasympathetic nervous system.

In a vagus nerve stimulation study of child patients with treatment-resistant epilepsy, the responders to this treatment showed greater increased anisotropy in the left limbic, thalamocortical, and association fibers, and they demonstrated increased functional connectivity in a network within the thalamus, temporal nodes, and left Ic [48].

These neuroimaging findings identified a substrate for central nervous system mechanisms that contribute to the sympathovagal balance of the Ic, suggesting that the right Ic has an association with sympathetic nervous system activity while the left Ic has an association with parasympathetic nervous system activity.

## 3. Stroke Involving the Insular Cortex and the Disrupted Cardiovascular System

### 3.1. Stroke Involving the Insular Cortex and Cerebrogenic Cardiac Arrhythmias

In the acute phase of a stroke such as a cerebral hemorrhage or subarachnoid hemorrhage (SAH), ECG changes have sometimes been observed, including corrected QT (QTc) interval prolongation, ST-segment elevation or depression, and T-wave inversion [49,50,51,52,53,54]. Inversions in the T-wave that were concomitant with higher concentrations of plasma norepinephrine were frequently reported within first 24 h after onset in patients with SAH [55]. Ic-involved stroke was reported to be independently associated with QT prolongation [56]. In patients with ischemic stroke involving the right Ic, an increased QTc interval was an independent predictor of poor prognosis at two years [57]. In a case report, acute right MCAO including Ic ischemia was associated with acute LV dysfunction, T-wave inversion, and QTc prolongation [58]. Acute lesion involving the Ic was reported to be associated with ST segment elevation, ectopic beats, and sinus tachycardia [59].

Patients with a parietoinsular ischemic lesion were observed to have an increased risk of developing atrial fibrillation detected after stroke (AFDAS) [60]. Although the pathophysiology of AFDAS is not clearly known, AFDAS is thought to have a higher incidence of stroke involving the Ic compared to AF that is known before stroke onset (KAF). The main objective of an ongoing systematic review [61] is to compare the proportions of stroke involving Ic, recurrent stroke, and deaths in patient groups with no AF, KAF, and AFDAS. However, in that systematic review, sufficient data has not yet been obtained to determine whether there are significant differences in stroke involving the Ic among no AF, KAF, and AFDAS [62].

### 3.2. Stroke Involving the Insular Cortex and Asystole/Disrupted Diurnal BP Variation/Reduced Baroreflex Sensitivity/Acute Kidney Injury

Asystole was reported in patients with ischemic stroke involving the right Ic [63] or right Ic hemorrhage [64]. Patients who had experienced an ischemic stroke involving the Ic more frequently showed a rise in nocturnal BP, and they presented higher norepinephrine levels compared to those of patients without Ic-involved ischemia [65]. In patients who had a first-ever cerebral infarction, a rise in nocturnal BP, a higher serum norepinephrine level, an infarction in the right hemisphere, and the presence of Ic infarction were independent predictors of unfavorable one-year functional outcomes [66]. The highest sympathetic nervous system activation was observed in patients with a stroke involving the right Ic [67]. Stroke patients with right Ic involvement were more prone to developing cardiovascular autonomic dysfunction. Patients with a left Ic-involved stroke had a significant decrease in baroreflex sensitivity (BRS) compared to those with a right Ic-involved stroke [68].

The role of an infarction involving the Ic in acute kidney injury (AKI) was analyzed in 172 acute ischemic stroke patients; the results demonstrated that Ic infarction, especially right Ic infarction, is an independent risk factor for AKI [19].

### 3.3. Stroke Involving the Insular Cortex and Cardiac Overload/Injury

In the Jichi Medical School ABPM Study Wave 2 Core, left Ic atrophy was significantly correlated with the level of brain natriuretic peptide (BNP) [4,69]. Stroke with Ic involvement was associated with an elevated serum level of N-terminal prohormone of BNP (NT-proBNP) [70], suggesting a relationship between Ic damage and cardiac overload or dysfunction [71]. Ic-involved stroke was also associated with increased cardiac troponin T (cTnT) [72,73], and Ic involvement and higher cTnT on admission were independently associated with the subsequent detection of AF [74].

Because it is not yet known which comorbidities and stroke characteristics are associated with elevated post-stroke cTnT levels, an ongoing meta-analysis is assessing the association between elevated cTnT and specific stroke characteristics such as infarct/hemorrhage size, stroke severity, Ic involvement, and renal failure after ischemic stroke or ICH [75].

Ic-involved stroke is suggested to be associated with changes in ECG, disrupted diurnal BP variation, reduced BRS, cardiac injury, cardiac overload, and cardiac dysfunction. Instability of the sympathovagal balance could serve as substantial pathophysiology linking the relationship between Ic lesion and dysregulation in the cardiovascular system. A meta-analysis examining the relationship between Ic-involved stroke and cardiac injury is ongoing, and a more clinically relevant pathophysiology of Ic-involved stroke can thus be expected to be revealed in relation to cardiovascular dysregulation.

## 4. Takotsubo Syndrome

TTS was first officially reported in 1991 as a reversible cardiomyopathy that appeared to be precipitated by acute emotional stress [1]. The majority of TTS patients were postmenopausal females and usually developed symptoms similar to those of an acute coronary syndrome [76,77]. A strong emotional stressor was thought to cause a transient abnormality of LV wall motion in the apical and mid-ventricular portion (Figure 2) without obstructive coronary artery disease (Figure 3) with ECG changes (Figure 4). TTS was initially called “Takotsubo cardiomyopathy” [1,78,79,80]. Although most TTS cases were described as having been caused by negative emotions [4], there are cases in which positive emotions, such as joy, triggered TTS [81,82]. In addition, 10% of TTS patients are male [83,84].

### 4.1. Pathophysiology of Takotsubo Syndrome

Sympathetic autonomic nervous system overactivity is one possible pathophysiology underlying TTS. Elevated concentrations of plasma norepinephrine have been reported in acute stroke [85,86], and ECG changes in TTS were also observed after an intravenous [87,88] or intracoronary [89] injection of catecholamine. Increased sympathetic nervous system activity is an apparent link in the relationship between cerebrovascular disease and TTS specifically in postmenopausal women.

Recent research has described a pathophysiology linking the relationship between cerebral infarction and TTS with poor outcome, and damage to the integrity of the blood–brain barrier (BBB) was suggested to explain the underlying pathophysiology. In addition to increased levels of catecholamine, post-cerebral infarction patients with TTS showed elevated blood levels of inflammatory mediators that are known to cause vascular conditions such as thromboembolism and stroke [90]. These stressors concomitant with hypoxia were shown to deteriorate BBB integrity, which could be the mechanism underlying the poor outcomes of Ic-involved stroke, as Ic neurons were also shown to encode and retrieve specific immune responses [91,92].

Using an in vitro model of the BBB consisting of an immortalized murine microvascular endothelial cell (cEND) line, Förster et al. (2005) [93] analyzed the molecular effects of exposure to catecholamines (dopamine, norepinephrine, and epinephrine), pro-inflammation cytokines (interleukin [IL]-6 and tumor necrosis factor-alpha [TNF-α]). They simultaneously subjected the cell system to oxygen glucose deprivation (OGD; which had been established as an in vitro cerebral infarction model) with and without subsequent reoxygenation [94]. Their results demonstrated that the BBB’s integrity and the cells’ morphology and viability were clearly affected by catecholamine and inflammation under the conditions of OGD. Most proteins of the established BBB model were downregulated (Figure 5). The structures affected could form the basis of the molecular pathophysiology of the cerebral vasculature and comprise a potential therapeutic target for cerebral infarction concomitant with TTS [95].

### 4.2. Case Report

We reported the TTS case of a 79-year-old female with LV asynergy in the mid-ventricle after a right Ic infarction (Figure 6). Twelve hours after her hospitalization, the patient suddenly collapsed. After rapid cardiopulmonary resuscitation, spontaneous circulation returned. Coronary angiography was then performed and showed no obstructive arterial disease (Figure 7). In left ventriculography, akinesis was observed in the mid-portion with hyperkinesis in the basal and apex portions of the LV (Figure 8). After two weeks of hospitalization, the LV contraction in the mid-portion was improved as shown by cardiac ultrasound. This patient’s right Ic infarction might have been associated with CAN dysregulation that was linked to an increase in sympathetic nervous system activity that triggered TTS [96].

In a study of 1750 TTS patients, the apical pattern of TTS was shown to be the most common type (81.7%), and the mid-ventricular pattern was the second most common type (14.6%) [97]. Although Ic lesion is considered to play a pivotal role as a trigger of TTS [4], a TTS patient with the mid-ventricular pattern had never been reported in relation to Ic damage until our above-described patient [96].

In the apical pattern of TTS, the effect of increased catecholamine levels on the myocardium determined by the beta-adrenoceptor (βAR) gradient is thought to be associated with subsequent acute apical ballooning [98]. βAR gradients in the LV might be different between the apical pattern and the mid-ventricular pattern of TTS. The murine cardiac β1AR function was shown to be suppressed by β2AR overexpression [99]; there was a bell-shaped dose-response relationship between the doses of epinephrine and myocardial contraction, and at the highest epinephrine doses, a negative inotropic effect was confirmed in relation to the β2AR. In our TTS patient’s case, the highest β2AR distribution might have been in the mid-portion of the LV.

### 4.3. Ic Stroke and Takotsubo Syndrome

Data from the Japanese National Cardiovascular Center [100] revealed that among 569 consecutive patients with acute cerebral infarction, seven patients developed TTS. All seven of these patients were female, and six were ≥75 years old. In these six patients, the culprit infarcts included the Ic or were identified close to the Ic. Elderly females with an Ic lesion were thus suggested as predominant features for the development of TTS among patients with acute ischemic stroke [100]. In a Korean hospital-based registry, the patients with both ischemic stroke and TTS also had a higher inflammatory marker level and a higher prevalence of lesions in the Ic and peri-Ic areas [101].

In a review of patients with acute stroke concomitant with the development of TTS [102], the female proportion was 77%, and the average age of the patients was 72.5 years. Stroke involving the Ic was found in 38.4% of the patients. The patients’ mean score on the U.S. National Institutes of Health Stroke Scale (NIHSS) was 12.6 at admission and 10.8 at discharge. ST-segment elevations and T-wave inversions were observed in 69.2% and 84.6% of the cases, respectively, whereas the average TnT was 0.64 mcg/dL. The average LV ejection fraction (LVEF) was 34.4% at the time of onset; LVEF was significantly improved in 84.6% of the patients at approximately four weeks after onset [102]. In addition, the majority of the patients presented with an apex-involved pattern [103].

## 5. Possible Role of Estrogen Deficiency in Takotsubo Syndrome

A decrease in the level of ovarian hormones has been shown to accelerate aging in women and increase their rate of mortality due to cardiovascular disease [104]. Greater attention has recently been paid to heart failure in post-menopausal women. In the Multi-Ethnic Study of Atherosclerosis (MESA), during 12.1 years of follow-up in 2834 women after menopause, a higher testosterone/estradiol ratio was significantly associated with the incidence of heart failure with a reduced EF (HFrEF) [105].

A study by Barros et al. (2006) [106] revealed that estradiol has an important role in the regulation of energy metabolism, and it has provided a new notion about the role of estrogen receptor alpha (ERα) and ERβ. ERs were shown to be expressed in the blood vessels [107], and multiple functional abnormalities were observed in vascular smooth muscle cells and blood vessels of ERβ-deficient mice [108]. Vasoconstriction via an ERβ-mediated increase in the expression of inducible nitric oxide synthase is attenuated by estrogen in wild-type mice. On the other hand, in blood vessels of ERβ-deficient mice, vasoconstriction is augmented by estrogen. With the increase in age, ERβ-deficient mice were observed to develop sustained systolic/diastolic hypertension [108]. These data indicate that ERβ has a pivotal role in the regulation of BP and vascular function.

Although it is known that the heart reacts to estrogen, it is not yet clear whether estrogen affects cardiomyocytes directly or indirectly via the immune, nervous, and/or vascular systems. In analyses of ERβ-deficient mice, enlarged hearts were observed with myocardial disarray, disrupted intercalated discs, profound nuclear structure alteration, and an increased number and size of gap junctions [109]. In the same study’s immunohistochemical analysis using ERβ antibodies, ERβ was not detected in the cardiac muscle, indicating that the abnormal cardiac morphology in the hearts of the ERβ-deficient mice might be derived from stress on the nuclear envelope due to a sustained pressure overload in systolic/diastolic BP [109]. The indirect effects of estrogen on the heart muscle thus seem to be supported, because ERs were not detected in myocardium [109]. Reports on the expression of ERα/β in cardiomyocytes are conflicting, underlining the possibility of indirect estradiol effects on the heart muscle [109,110].

Although the clinical features of HFrEF and those of TTS differ, the postmenopausal female status might serve as a common pathophysiology underlying HFrEF and TTS, and a reduced level of estradiol seems to be a common risk. The complex relationships between sex hormones and the cardiovascular regulatory system might give us a glimpse into the pathophysiology of HFrEF and TTS.

## 6. Sex-Specific Neuroanatomy of Takotsubo Syndrome: How Does the Ic Interact?

### 6.1. Estrogen Deficiency and Functional Cerebral Asymmetry in TTS

The right hemisphere is predominantly involved in controlling spontaneously triggered emotional stimuli [111], and it shares a final common path for processing emotional information [112]. Studies of the influence of estrogen on cardiac sympathovagal modulation and on asymmetry of cerebral function have indicated that estrogen has a positive correlation with activation of the left hemisphere [5], and a link between a decrease in the endogenous estrogen level and inactivation of the left hemisphere was suggested. A lower estrogen level might therefore shift the sympathovagal balance to an increase in sympathetic nervous system activity due to an inactivation of the left hemisphere, since the left hemisphere has a predominant association with cardiac parasympathetic tone [5].

### 6.2. Circuitry and Functional Aspects of the Ic in Takotsubo Syndrome

The Ic has abundant reciprocal connections with brain regions in the frontal, temporal, and parietal lobes as well as in the thalamus and the limbic/paralimbic systems [113,114,115]. Dense reciprocal connections are observed between the Ic and subcortical autonomic centers of the NTS [116,117], the PBN [118], and the lateral hypothalamic area (LHA) [119], and these autonomic core centers also have reciprocal connections with each other.

The Ic is involved in processing the perception of positive as well as negative emotional information. The above-described evidence supports the notion that the exaggerated right Ic activation in anxiety, fear, phobia, and happiness experienced by patients with TTS changes the predominant sympathovagal balance to cardiac sympathetic overdrive. In postmenopausal females, left-hemispheric reduced activation including the Ic shifts the sympathovagal balance to sympathetic predominance, and this might be associated with a pivotal role of the Ic in the pathophysiology of TTS in relation to positive and negative emotional events.

It is not yet known whether the transient LV dysfunction observed in neurological disease has the same pathophysiology as conventional TTS that is free from overt neurological disorders. A lesion located at or close to the Ic may help explain the crucial role of sympathetic overactivation in post-menopausal women.

In a resting-state fMRI study, TTS patients were observed to have reduced functional connectivity in sympathovagal-associated subnetworks including the bilateral medial orbitofrontal cortices, the left posterior cingulate cortex, and the left anterior Ic [120]. Because the left Ic is associated with parasympathetic nervous system activity, it is possible that the hypoconnected subnetwork including the left Ic triggers TTS due to the shift in the sympathovagal balance to relatively higher sympathetic nervous system activity.

Resting-state and anatomical fMRI studies demonstrated functional and volumetric right Ic changes in TTS patients, and it was shown that in TTS, the relationship between left amygdala atrophy and right anterior Ic atrophy was associated with sympathovagal instability [121]. In the JMS ABPM Study (Wave 2 Core), only the volume of the right Ic was significantly negatively correlated with the serum noradrenaline level [4,69].

Animal models of experimental stroke revealed that right-side MCAO was associated with neurochemical derangements in the amygdala and Ic of the ipsilateral hemisphere, which led to enhanced cardiac sympathetic outflow with an increased level of synaptic norepinephrine [122]. Increased sympathetic outflow can also occur due to a cerebral infarction, which could induce stress cardiomyopathy [80]. An fMRI study demonstrated that the lateralized salience network integrity of the right amygdala for the sympathetic nervous system and that of the left Ic for the parasympathetic nervous system were associated with sympathovagal stability [123]. When an individual confronts danger, a combination of autonomic, behavioral, and cognitive responses occurs for self-preservation. These features are shown to be merged at the Ic, which fine-tunes behavioral responses to match the level of expected danger [124,125]. In light of these findings, it is possible that altered amygdalo-insular connectivity might also play a pivotal role as a TTS trigger, linked to increased sympathetic nervous system activity (Figure 9).

## 7. Conclusions

Recent research concerning the relationship between the Ic and autonomic control of the cardiovascular system has been summarized and updated in this review. Stroke involving the Ic presents various types of cardio-autonomic dysregulation that are also observed in TTS. Left Ic dysfunction, which is associated with estrogen deficiency in postmenopausal status, might have a pivotal role in the sympathetic overdrive in TTS. Future research into the sex-specific neuroanatomy of the Ic in TTS may provide key insights into the pathophysiology of stress-related cardiovascular disease from a psychosomatic perspective.

## Figures and Tables

**Figure 1 biomolecules-12-00110-f001:**
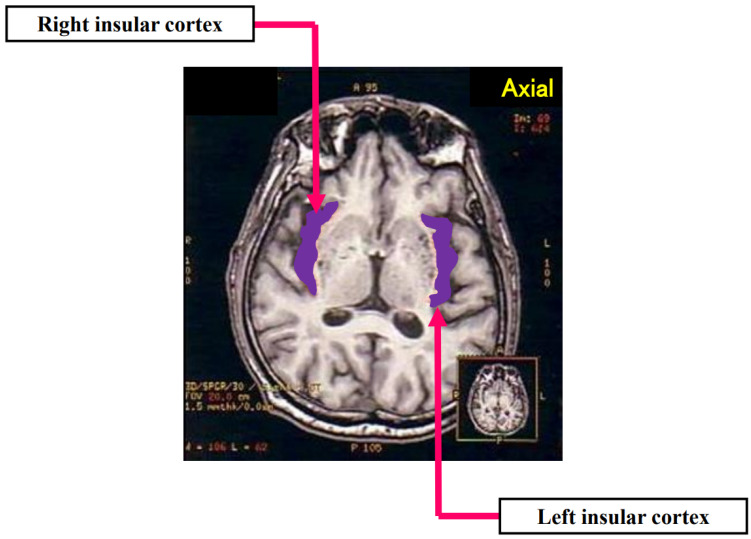
Axial T1-weighted MRI. White matter of the orbitofrontal cortex continues to the extreme capsule deep to the anterior insular cortex. Reconstructed from [4].

**Figure 2 biomolecules-12-00110-f002:**
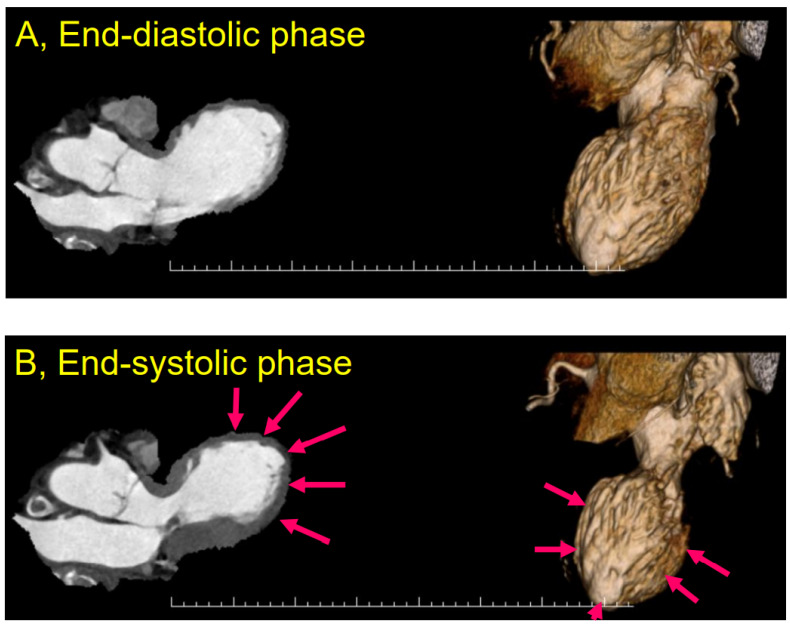
Left ventricular images assessed by computed tomography (CT) in Takotsubo syndrome (TTS). End-diastolic-phase left ventriculogram (**A**) and end-systolic-phase left ventriculogram (**B**). The extensive area around the apex portion shows akinesis (arrows), and the basal segments display hypercontraction, especially in the end-systolic phase (**B**). From [82].

**Figure 3 biomolecules-12-00110-f003:**
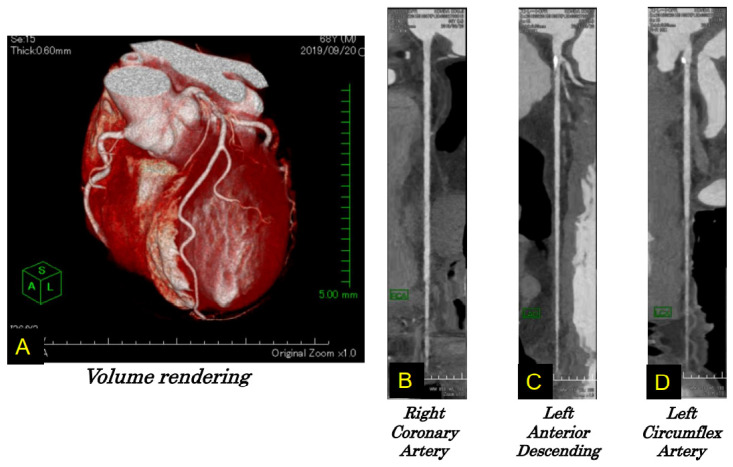
Coronary artery assessed by CT. In the volume rendering image (**A**), three vessels were observed. No stenosis can be seen in the right (**B**), left anterior descending (**C**), or left circumflex arteries (**D**). Reconstructed from [82].

**Figure 4 biomolecules-12-00110-f004:**
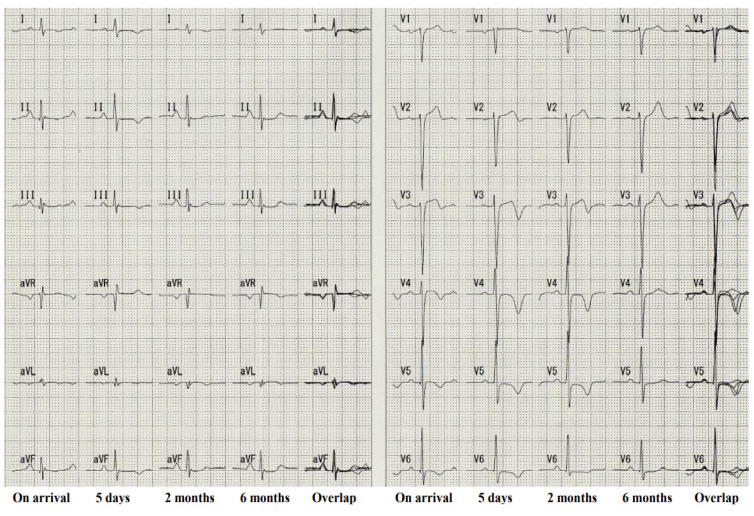
The time course of ECG changes from the patient’s admission to the 6-month follow-up. The overlap ECG waveform in each lead is presented on the right. Reconstructed from [82].

**Figure 5 biomolecules-12-00110-f005:**

Loss of morphological integrity of cEND cells exposed to elevated catecholamine levels and inflammatory mediators. Immunofluorescence staining of tight junction proteins claudin-5 (green) and ZO-1 (red) as markers of morphological changes of the endothelial cell monolayer. DAPI (blue) was used to stain nuclei. Cells were grown on cover slips to confluence. After differentiation, cEND cells were treated with a combination of catecholamines and inflammatory mediators (stress factors). Stress factors were administered under different incubation conditions. Magnification 400×, scale bar 20 μm. (**A**) Stress factor application for 4 h under normoxia conditions (4 h NORMOX). (**B**) Stress factor application for 24 h under normoxia conditions (24 h NORMOX). (**C**). Stress factor application for 4 h under oxygen glucose deprivation (OGD) conditions. (**D**) Stress factor application for 4 h under OGD conditions with 20 h of subsequent reoxygenation under normoxia conditions (REOX). (**E**) Cell number in treatments shown in panels (**A**–**D**) normalized to control. *** *p* < 0.001, **** *p* < 0.0001. From [95].

**Figure 6 biomolecules-12-00110-f006:**
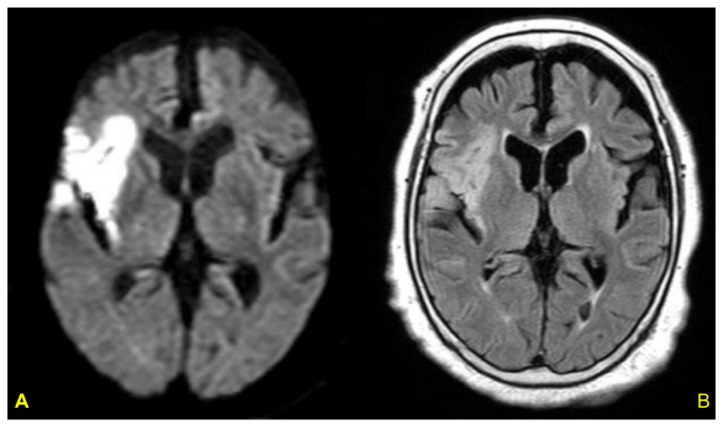
Axial brain MRI in a patient with TTS. Acute cerebral lesions including the right Ic were observed in the diffusion-weighted (**A**) and FLAIR (**B**) images. From [96].

**Figure 7 biomolecules-12-00110-f007:**
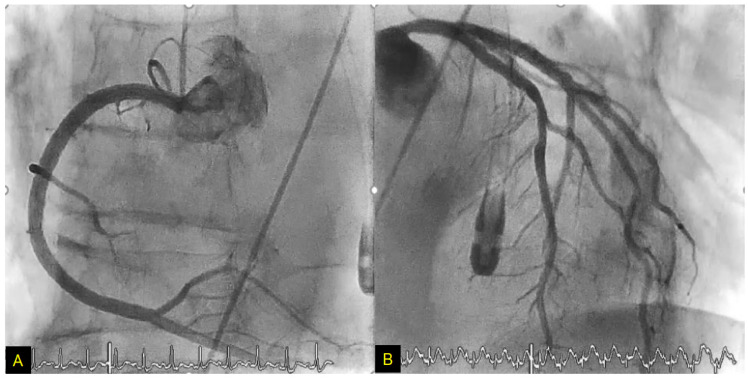
Coronary artery angiography. No stenosis was observed in the right coronary artery (**A**), left anterior descending artery, or circumflex artery (**B**). From [96].

**Figure 8 biomolecules-12-00110-f008:**
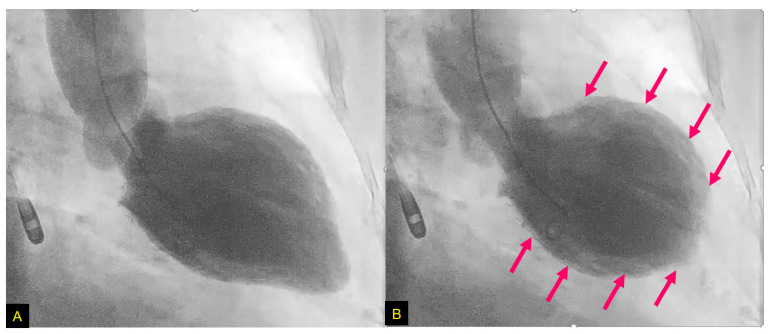
Left ventriculography. End-diastolic-phase left ventriculogram (**A**) and end-systolic-phase left ventriculogram (**B**). The extensive area around the mid-portion shows akinesis (arrows), and the basal and apex portions display hypercontraction, especially in the end-systolic phase (**B**). From [96].

**Figure 9 biomolecules-12-00110-f009:**
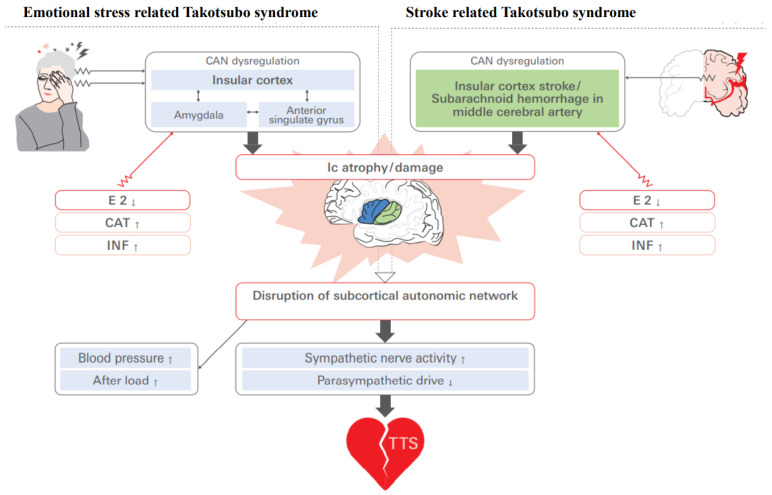
Possible pathways involving the Ic in the development of TTS in relation to emotional stressors and stroke. CAN: central autonomic network, E2: estradiol, CAT: catecholamine, INF: inflammatory cytokines.

## Data Availability

The data that support the findings of the study regarding this manuscript are available after the corresponding author’s approval upon reasonable request.

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
