# Peer review of "Sex Hormone-Specific Neuroanatomy of Takotsubo Syndrome: Is the Insular Cortex a Moderator?"

_biomolecules, 2022, doi:10.3390/biom12010110_

Round 1

Reviewer 1 Report

I don't have any further comments.

Reviewer 2 Report

Dear Authors

Thank you for considering all my comments in the revision of the manuscript.

I believe you have misunderstood my statement about copying figures from "other authors' works". I know that all figures come from other earlier works by the authors of the reviewed manuscript.

I found the article very interesting and had recommended the approval of the manuscript before. I still hold my opinion.

Best regards 

This manuscript is a resubmission of an earlier submission. The following is a list of the peer review reports and author responses from that submission.

Round 1

Reviewer 1 Report

Dear Authors

The pathophysiology of Takotsubo syndrome is not fully explained, so the subject of the presented review entitled "Sex-specific neuroanatomy of Takotsubo cardiomyopathy: How does the insular cortex interact?" it is up-to-date, interesting and the article is valuable.

I suggest using the current name "Takotsubo Syndrome" in the title.

Figures in the manuscript are from other authors' works but have been inaccurately copied:

Figure 2 A and B - cropped photos

Figure 3 - Where are B, C and D?

Figure 4 - missing aVF and V6 leads

Figure 5 - Where is D ?, E is too small

Figure 6 - cropped photo

Figure 7 - cropped photos Figure

8 - cropped photos Figure

9 - cut figure Figures are of great value in this review, so I propose to correct and organize them to improve the appearance of the manuscript. 

Response to the Reviewer#1

We are very grateful for your comments, which have substantially improved our manuscript. We considered the following points and summarized our responses and changes as follows. Text in the manuscript that has been revised is in blue font.

Major comments

  1. Comment: The pathophysiology of Takotsubo syndrome is not fully explained, so the subject of the presented review entitled "Sex-specific neuroanatomy of Takotsubo cardiomyopathy: How does the insular cortex interact?" it is up-to-date, interesting and the article is valuable.

Response: Thank you. We believe that this review provides some new insight of clinical and pathophysiological relevance.

  1. Comment: I suggest using the current name "Takotsubo Syndrome" in the title.

Response: We have changed to the term "Takotsubo Syndrome" in the title.

  1. Comment: Figures in the manuscript are from other authors' works but have been inaccurately copied:

Response: All of the figures were derived from our manuscripts. Please indicate which one is from other authors' works? As the reviewer pointed out, Fig.1 is not from Nagai et al., 2009 but from Nagai et al., 2017.

In the “2. The brain-heart connection 2.1. Cortical regulation in the circulatory system in response to emotional stress: The central autonomic network (CAN)”, we change: “The CAN functions as an integral component of an internal regulation system with which the brain controls the visceromotor and neuroendocrine systems (Benarroch, 1993).” to “The CAN functions as an integral component of an internal regulation system with which the brain controls the visceromotor and neuroendocrine systems (Benarroch, 1993; Nagai et al., 2017).”

In the Fig. legend for Fig.1, we changed: “Axial T1-weighted MRI. White matter of the orbitofrontal cortex continues to the extreme capsule deep into the anterior insular cortex. Reconstructed from Nagai et al. (2009).” to “Axial T1-weighted MRI. White matter of the orbitofrontal cortex continues to the extreme capsule deep into the anterior insular cortex. Reconstructed from Nagai et al. (2017).”

  1. Comment: Figure 2 A and B - cropped photos

Response: The reviewer is right. In the Fig. legend for Fig.2, we changed: “Reconstructed from Nagai et al. (2020)” to “From Nagai et al. (2020)”.

In the Fig. legend for Fig.2, we changed: “Reconstructed from Nagai et al. (2020)” to “From Nagai et al. (2020)”.

  1. Comment: Figure 3 - Where are B, C and D?

Response: Please see the red arrows below.

  1. Comment: Figure 4 - missing aVF and V6 leads

Response: Please see the red arrow (aVF) and the blue arrow (V6) below.

  1. Comment: Figure 5 - Where is D ?, E is too small

Response: Please see the red arrow (D) below.

We changed the composition of the figure E.

  1. Comment: Figure 6 - cropped photos

Response: The reviewer is right. In the Fig. legend for Fig.6, we changed: “Reconstructed from Osawa et al. (2021)” to “From Osawa et al. (2021)”.

In the Fig. legend for Fig.6, we changed: “Reconstructed from Osawa et al. (2021)” to “From Osawa et al. (2021)”.

  1. Comment: Figure 7 - cropped photos

Response: The reviewer is right. In the Fig. legend for Fig.7, we changed: “Reconstructed from Osawa et al. (2021)” to “From Osawa et al. (2021)”.

In the Fig. legend for Fig.7, we changed: “Reconstructed from Osawa et al. (2021)” to “From Osawa et al. (2021)”.

  1. Comment: Figure 8 - cropped photos

Response: The reviewer is right. In the Fig. legend for Fig.8, we changed: “Reconstructed from Osawa et al. (2021)” to “From Osawa et al. (2021)”.

In the Fig. legend for Fig.8, we changed: “Reconstructed from Osawa et al. (2021)” to “From Osawa et al. (2021)”.

  1. Comment: Cut figure Figures are of great value in this review, so I propose to correct and organize them to improve the appearance of the manuscript.

Response: As the reviewer suggested, we have corrected and organized Figs to improve the appearance of the manuscript.

Reviewer 2 Report

The review discusses potential link between Takotsubo cardiomyopathy, gender and neuroanatomy. Please consider my comments below.

  1. Figures are defined as reconstructions from already published projects, although the name of license (e.g. CC-BY) or agreement coming from both all co-authors and publisher are missing.
  2. Unappreciated extensive SELF-CITATION (!!!) – the number of author’s articles reached 14% of total number of references. Furthermore, the original data in most cases comes from previous author’s publications, while the other references seems to be used only as a “statistical background” enriching the total number of references.
  3. “The Ic is thus suspected to play a pivotal role in the regulation of the autonomic nervous system in relation to emotional significance (Nagai et al., 2010).” – this sentence includes the final conclusion, but does not explain the experimental background leading to such conclusion.
  4. Section 2.3.: In vivo studies include animal MALE-ONLY models, which remains in contract with working hypothesis stained in sections above. Indeed, authors underlined the role of female estrogens, while in vivo projects included male animals. It is not clear how it looks for in human projects, I might only assumed 50/50 (f/m) statistic.
  5. Figure 5 is incomplete.

Actually, within the MS, the authors fail to prove the working hypothesis. The review rather shows the epidemiology of the Takotsubo Syndrom rather than explains the critical role of female hormones in the Takotsubo Syndrom regulation. Furthermore, the idea based on observations going from clinical practice, and not very much from further scientific evaluation.

Response to the Reviewer#2

The similarity index has been reduced from 61% to 13%, and we have omitted a number of references of ourselves.

We are very grateful for your comments, which have substantially improved our manuscript. We considered the following points and summarized our responses and changes as follows. Text in the manuscript that has been revised is in blue font.

Major comments

  1. Comment: Figures are defined as reconstructions from already published projects, although the name of license (e.g. CC-BY) or agreement coming from both all co-authors and publisher are missing.

Response: This is very important issue. Fig1, 2 and 4 were reconstructed. And Fig. 2 was from Nagai, et al., 2020 (for see the attachment PDF for License Number 5213640918995). Fig.5, 6, 7, and 8 were free from permission. Author Agreement Form was also attached.

  1. Comment: Unappreciated extensive SELF-CITATION (!!!) – the number of author’s articles reached 14% of total number of references. Furthermore, the original data in most cases comes from previous author’s publications, while the other references seems to be used only as a “statistical background” enriching the total number of references.

Response: We have omitted a number of references of ourselves. The present review describes the sex-specific neuroanatomy in TTS and updates the knowledge regarding the integrated roles of the Ic in the brain–heart axis, not for a “statistical background” enriching the total number of references.

  1. Comment: “The Ic is thus suspected to play a pivotal role in the regulation of the autonomic nervous system in relation to emotional significance (Nagai et al., 2010).” – this sentence includes the final conclusion, but does not explain the experimental background leading to such conclusion.

Response: Klein et al, found that the insular cortex acts as a state-dependent regulator of fear that is necessary to establish an equilibrium between the extinction and maintenance of fear memories in mice. Whereas insular cortex responsiveness to fear-evoking cues increased with their certainty to predict harm, this activity was attenuated through negative bodily feedback that arose from heart rate decelerations during freezing. Perturbation of body-brain communication by vagus nerve stimulation disrupted the balance between fear extinction and maintenance similar to insular cortex inhibition. The insular cortex integrates predictive sensory and interoceptive signals to provide graded and bidirectional teaching signals that gate fear extinction and illustrate how bodily feedback signals are used to maintain fear within a functional equilibrium (Christianson, 2021; Klein et al., 2021).

 In the last sentence on “6.2. Circuitry and functional aspects of the Ic in Takotsubo syndrome”, we added: “When an individual confronts danger, a combination of autonomic, behavioral, and cognitive responses occurs for self-preservation. These features are shown to be merged at the Ic, which fine-tunes behavioral responses to match the level of expected danger (Christianson, 2021; Klein et al., 2021).”

Christianson, J.P. (2021). The head and the heart of fear. Science. 374, 937-938.

Klein, A.S., Dolensek, N., Weiand, C., and Gogolla, N. (2021). Fear balance is maintained by bodily feedback to the insular cortex in mice. Science 374, 1010-1015.

  1. Comment: Section 2.3.: In vivo studies include animal MALE-ONLY models, which remains in contract with working hypothesis stained in sections above. Indeed, authors underlined the role of female estrogens, while in vivo projects included male animals. It is not clear how it looks for in human projects, I might only assumed 50/50 (f/m) statistic.

Response: The reviewer is right. These studies were not focused on the gender effect. This would be the limitation to interpret the animal model for the TTS.

 In the last paragraph on “A. Animal studies” in “2.3. The insular cortex and the regulation of the cardiovascular system”, we added: “Because there were few observations from the viewpoint of gender differences, in the future, it would be necessary to examine the gender-specific brain-heart relationship in these animal experiments.”

  1. Comment: Figure 5 is incomplete.

Response: Yes, we changed the layout of Fig.5.

  1. Comment: Actually, within the MS, the authors fail to prove the working hypothesis. The review rather shows the epidemiology of the Takotsubo Syndrom rather than explains the critical role of female hormones in the Takotsubo Syndrom regulation. Furthermore, the idea based on observations going from clinical practice, and not very much from further scientific evaluation.

Response: Although we could understand what the reviewer was presenting to some extent, there have been rarely provided the importance of female hormones on trigging TTS in the literatures. Thus, this review article would provide some new insight into the study area regarding TTS.